# Shared Protentions in Multi-Agent Active Inference

**DOI:** 10.3390/e26040303

**Published:** 2024-03-29

**Authors:** Mahault Albarracin, Riddhi J. Pitliya, Toby St. Clere Smithe, Daniel Ari Friedman, Karl Friston, Maxwell J. D. Ramstead

**Affiliations:** 1VERSES Research Lab and Spatial Web Foundation, Los Angeles, CA 90016, USA; riddhi.jain@verses.ai (R.J.P.); entropy@tsmithe.net (T.S.C.S.); k.friston@ucl.ac.uk (K.F.); maxwell.ramstead@verses.ai (M.J.D.R.); 2Département d’Informatique, l’Université du Québec à Montréal, Montreal, QC H3C 3P8, Canada; 3Department of Experimental Psychology, University of Oxford, Oxford OX2 6GG, UK; 4Topos Institute, Berkeley, CA 94704, USA; 5Active Inference Institute, Davis, CA 95616, USA; daniel@activeinference.institute; 6Wellcome Trust Centre for Neuroimaging, Institute of Neurology, University College London, London WC1N 3BG, UK

**Keywords:** active inference, phenomenology, multi-agent, category theory

## Abstract

In this paper, we unite concepts from Husserlian phenomenology, the active inference framework in theoretical biology, and category theory in mathematics to develop a comprehensive framework for understanding social action premised on shared goals. We begin with an overview of Husserlian phenomenology, focusing on aspects of inner time-consciousness, namely, retention, primal impression, and protention. We then review active inference as a formal approach to modeling agent behavior based on variational (approximate Bayesian) inference. Expanding upon Husserl’s model of time consciousness, we consider collective goal-directed behavior, emphasizing shared protentions among agents and their connection to the shared generative models of active inference. This integrated framework aims to formalize shared goals in terms of shared protentions, and thereby shed light on the emergence of group intentionality. Building on this foundation, we incorporate mathematical tools from category theory, in particular, sheaf and topos theory, to furnish a mathematical image of individual and group interactions within a stochastic environment. Specifically, we employ morphisms between polynomial representations of individual agent models, allowing predictions not only of their own behaviors but also those of other agents and environmental responses. Sheaf and topos theory facilitates the construction of coherent agent worldviews and provides a way of representing consensus or shared understanding. We explore the emergence of shared protentions, bridging the phenomenology of temporal structure, multi-agent active inference systems, and category theory. Shared protentions are highlighted as pivotal for coordination and achieving common objectives. We conclude by acknowledging the intricacies stemming from stochastic systems and uncertainties in realizing shared goals.

## 1. Introduction

This paper proposes to understand collective action driven by shared goals by formalizing core concepts from phenomenological philosophy—notably Husserl’s phenomenological descriptions of the consciousness of inner time—using mathematical tools from category theory under the active inference approach to theoretical biology. This project falls under the rubric of computational phenomenology [1] and pursues initial work [2,3] that proposed an active inference version of (core aspects of) Husserl’s phenomenology. Our specific contribution in this paper will be to extend the core aspects of Husserl’s description of time consciousness to group action and to propose a formalization of this extension.

In detail, we unpack the notion of shared goals in a social group by appealing to the construct of protention (or real-time, implicit anticipation) in Husserlian phenomenology. We propose that individual-scale protentions can be communicated (explicitly or implicitly) to other members of a social group, and we argue that, when properly augmented with tools from category theory, the active inference framework allows us to model the resulting shared protentions formally in terms of a shared generative model. To account for multiple agents in a shared environment, we extend our model to represent the interaction of agents having different perspectives on the social world, enabling us to model agents that predict behavior—both their own and that of their companions—as well as the environment’s response to their actions. We utilize sheaf-theoretic and topos-theoretic tools from category theory to construct coherent representations of the world from the perspectives of multiple agents, with a focus on creating “internal universes” (topoi) that represent the beliefs, perceptions, and predictions of each agent. In this setting, shared protentions are an emergent property—and possibly a necessary property—of any collective scale of self-organization, i.e., self-organization where elements or members of an ensemble co-organize themselves.

This paper weaves together elements that might seem relevant to fairly disparate readerships, namely, Husserlian phenomenology, active inference modeling, and category theory. We clarify that our primary intended audience is threefold. The primary intended audience is composed, in part, of phenomenologists who are interested in using contemporary mathematical approaches to generate formal models of the kinds of dynamic lived experiences that are captured by phenomenological descriptions. This segment of our readership will likely intersect with proponents of the project to naturalize phenomenology [4,5]. Our target audience also comprises active inference modelers who have taken an interest in consciousness and phenomenological description.

We begin by reviewing aspects of Husserlian temporal phenomenology, with a particular focus on the notions of primal impression, retention, and protention. These provide us with a conceptual foundation to think scientifically about the phenomenology of the emergence of shared goals in a community of interacting agents. We cast these shared goals in terms of the protentional (or future-oriented) aspects of immediate phenomenological experience, in particular, what we call “shared protentional goals”. We then formalize this neo-Husserlian construct with active inference, which allows for the representation and analysis of oneself and another’s generative models and their interactions with their environment. By using tools from category theory, namely, polynomial morphisms and hom polynomials, we are able to design agent architectures that implement a form of recursive cognition and prediction of other agents’ actions and environmental responses. Finally, we propose a method for gluing together the internal universes of multiple agents using topoi from category theory, allowing for a more robust representation and analysis of individual and group interactions within a stochastic environment. We use these tools to construct what we call a “consensus topos”, which represents the understanding of the world that is shared among the agents. This consensus topos may be considered the mathematical object representing the external world, providing a unified framework for analyzing social action based on shared goals. Our integrative approach provides some key first steps towards a computational phenomenology of collective action under a shared goal, which may help us naturalize group intentionality more generally and better understand the complex dynamics of social action.

## 2. From the Phenomenology of Time Consciousness to Co-Construction and Shared Goals

### 2.1. Overview of Husserlian Phenomenology of Inner Time-Consciousness

This section provides an examination of the intricate (but informal) descriptive study of the conscious perception of internal time that was proposed by the originator of the discipline of phenomenology, Husserl [1,6,7,8]. In philosophy, the term “phenomenology” is used in a technical sense to denote a specific kind of philosophical discourse, namely, the descriptive study of the dynamics, structure, and contents of the first-person, conscious experience. (The term is also used, less formally, to denote the descriptions that result from such an exercise and, more generally, to denote the way in which something discloses itself to a conscious subject.) Husserl himself defined phenomenology as a systematic effort to offer precise descriptions of the essential or necessary properties that pertain to various kinds of first-person experiences, by virtue of them being the kinds of experiences that they are. Thus, we can think of phenomenology as an informal counterpart to mathematics that is interested in the “essence” or essential properties of certain kinds of experience. Here, we focus on Husserl’s description of *inner time consciousness* and interpretations thereof, keeping in mind, of course, that Husserl scholarship is an active field of philosophical research and that any given interpretation will be subject to dispute. In what follows, we mainly draw from Husserl’s early lectures on time consciousness [6] and intersubjectivity [9] and on formal approaches to Husserl [2,3].

We appeal to Husserl’s phenomenological descriptions of inner time-consciousness over those of other phenomenologists, e.g., de Beauvoir or Merleau-Ponty, for a number of key reasons. The first is practical: Husserl’s extensive body of work provides us with what are arguably the most comprehensive, rigorously conducted, and rich descriptions of first-person experience available in the phenomenological literature, which, in addition, are perhaps the most amenable to mathematical formalization, as Husserl himself attempted to do in a few key places. Although others in the phenomenological tradition, such as Heidegger, have proposed descriptions of time consciousness, we would argue that they are neither as descriptively rich as Husserl’s nor as amenable to mathematization. Second, we chose Husserl’s phenomenological descriptions as a starting point because the recent project of computational phenomenology has already been developed by formalizing Husserl’s descriptions of inner time-consciousness using active inference modeling. So, we chose Husserl’s descriptions in part to build on and make the most of previous work. We should note that, following the tradition in naturalized phenomenology (e.g., [5]), we propose to mainly use Husserl’s phenomenological descriptions to generate data to be explained using generative modeling and bracket his antinaturalist philosophical commitments (see [4] for a discussion).

Husserl argues that the consciousness of inner time is the fundamental form of consciousness, acting as the background against which all other forms of conscious experience are situated and unfold. What Husserl calls the “constitution” of objects of experience (i.e., their disclosure to a perceiving subject) always presupposes the consciousness of time as a background condition [6,10,11]. Crucially, much like other thinkers of the time, like Bergson [12] and James [13], Husserl observes that the consciousness of time exhibits a form of “temporal thickness”. In this view, the way in which objects are experienced always evinces a kind of temporal depth. That is, any given experiential “now” carries with it a dimension of the just-passed and the just-yet-to-come. Husserl posits that the stream of conscious experience consists mostly of raw sensations or sensory data that arise in a basic, unprocessed state; these are called “primal impressions.” Husserl refers to this primal stratum of sensory experience as the “hyletic” data of consciousness (derived from the Greek term for matter, *hyle*) [6]. These data are then “formatted”, so to speak, in accordance with the cognitive principles governing the awareness of internal time.

In this context, “retention” refers to the aspect of time awareness that preserves the previous state or path of the temporal object in a particular manner: as “living” and contributing to the present “now” of perception. Husserl describes retention using the metaphor of “sediment” that accumulates over time. What Husserl calls “protention”, in contrast, refers to the element of time consciousness that looks ahead, so to speak, and anticipates the immediate future state or path of a temporal object. Retention and protention are both distinguished from the explicit recollection of a past event and the explicit imagining of a possible future event.

Husserl contends that our experience of temporally extended objects consists of a flow of anticipation (via protention) and fulfillment/frustration by new primal impressions, which may or may not conform with what was predicted to happen. Our inner time-consciousness consists of a dynamic process that anticipates what will be experienced next based on what has just been experienced. Thus, the flow of time consciousness is composed of a series of structured impressions. Primal impression, retention, and protention interact to shape the temporal flow of conscious experience. Retentions and protentions create a framework of sorts that structures the ebb and flow of conscious experience, which in turn affects how we perceive and anticipate events. The temporal thickness of experience, in this view, consists of the protented aspects of experience interacting with “sedimented” retentions.

### 2.2. Shared Protentions

Through the analysis of the phenomenological components of temporal consciousness, we can gain a valuable understanding of how individuals develop collective objectives and expectations and how these are incorporated into the previously mentioned mathematical models. Indeed, missing from the above account are *intersubjectivity* and the *sharing* of goals by agents in a shared life world. Husserl himself devoted much energy to thinking about intersubjectivity [9]. Zooming out to the broader literature, the concept of sharing goals or beliefs can be understood through several interrelated perspectives. Group members may possess shared beliefs about the past as well as aligned expectations about the future that are not explicitly expressed but that naturally coincide due to shared experiences or comprehension [14]. We would equate these shared ways of anticipating and smoothly coping with the world as being premised on a set of “shared protentions”.

We now make a terminological clarification. This paper concerns what we will call, in a somewhat idiosyncratic fashion, “shared protentions”. It should be noted that, for Husserl, in some sense, the structure of time consciousness, as an intentional relation between primary content, retention, and protentions, is shared by all conscious subjects: the structure of inner time-consciousness is invariant and identically the same for all conscious subjects. That is, according to Husserl, inner time-consciousness must conform to this structure by virtue of being the experience of the type inner time-consciousness.

The co-constitution of the lived world (Lebenswelt) has been discussed extensively by Husserl and studied under the rubric of intersubjectivity by phenomenologists of various sorts. Husserl distinguishes between the empirical and transcendental self or ego. For Husserl, the self generically functions as one of two “poles” of conscious experience (the “act-pole” and the “object-pole”). The transcendental self is, on the one hand, a kind of abstract structure that any conscious experience contains intrinsically. The self is also a thing that appears to this transcendental self; this is the empirical or personal self, which also has a temporal structure and accumulates experiences, forming habits. This empirical self is not just a static center of acts but evolves over time, integrating past acts into a cohesive identity. In his later writings, in particular in the fifth Cartesian Meditation [15], The Crisis of European Sciences and Transcendental Philosophy [16], and the extensive notes On Intersubjectivity, Husserl extends his analysis of the self, considered through the lens of intersubjectivity.

In Husserl’s account, the empirical self is co-constituted with others, where past experiences become integrated into the self. Since these experiences almost always involve other selves, their perspectives are thus integrated to the self. The self is thus a collective achievement that emerges from a community of selves [17]. The self realizes its constitutive role only within a network of intersubjective relations to other selves, implying an intrinsic connection between individual consciousnesses. The self is co-constructed through this intersubjective framework, where each self intentionally carries within itself the presence of other selves, thereby forming a deep, communicative relationship that establishes the full sense of the world. This communal interaction and the acknowledgment of each other’s subjective experiences contribute to the continuous generation and reformation of the self, emphasizing a dynamic, interrelated construction of identity and understanding. Our individual temporal experiences, the immediate experience of the now, retention, and protention, are not closed off to ourselves. They are open and connected to the temporal experiences of others. This openness allows for a shared temporal framework, or “world-time”, that encompasses not only our own temporal flow but also the temporal experiences of others, making our individual sense of time inherently intersubjective [18].

In the flow of time consciousness, the contents of experience become “formatted” by retention and protention, as pure forms of time consciousness, as they well up. It is these sedimented contents to which we refer—as shorthand—as “shared retentions” and “shared protentions”, in the sense that the content is shared between conscious subjects.

Previous work on modeling Husserlian time consciousness with active inference has associated these contents that are retained and protended by the pure protentional and retentional structure of inner time-consciousness with the implicit knowledge that is encoded in the parameters of a generative model, which captures the formal structure of inference. This is to be contrasted with the kind of online, contentful inference that corresponds to posterior state estimation in real time, i.e., with explicit predictions about the future. Thus, we should note that protentions are not equivalent to explicit predictions (neither in Husserl’s sense of formal structure nor in our sense of retained or protended contents). The point we are making is that shared intentionality depends on shared sedimented content; that is, in addition to having in common the generic form of inner time-consciousness, shared protentional content is necessary for shared intentionality.

Assuming the plausibility of such a neo-Husserlian concept, group members can be seen as actively and implicitly aligning their beliefs and expectations through dialogue and interaction, thereby enhancing their ability to predict each other’s actions and intentions [19], and thereby coming to perceive and act in the world in similar ways. Thus, shared retentions and protentions might arise collectively within the group as shared styles of appraising and engaging with the social world, extending beyond the individual agent’s subjective experience and leading to a shared implicit comprehension that surpasses and encompasses individual viewpoints.

In the following analysis, we propose to formally model these shared temporal structures, which involve resonant cognitive models and communication and which impact the decision-making and behavior of agents within a social group.

## 3. An Overview of Active Inference

Here, we provide a brief overview of active inference, oriented towards application in the setting of shared protentional goals (for a more complete overview of active inference, see [20,21,22], and for its application to modeling phenomenological experience, also known as “computational phenomenology”, see [23]).

Active inference is a mathematical account of the behavior of cognitive agents, modeling the action–perception loop in terms of variational (approximate Bayesian) inference. A generative model is defined, encompassing beliefs about the relationship between (unobservable) causes *s*—whose temporal transitions depend upon action *u*—and (observable) effects *o*, formalized as [24]
$$P\left(o_{\tau },s_{\tau },u_{\tau }|s_{\tau -1}\right)=\underset{likelihood}{\underset{︸}{P(o_{\tau }|s_{\tau })}}\underset{prior}{\underset{︸}{P\left(s_{\tau }|s_{\tau -1},u_{\tau }\right)P\left(u_{\tau }\right)}}$$

This probabilistic model contains a Markov blanket in virtue of certain conditional independencies—implicit in the above factorization—that individuate the observer from the observed (e.g., the agent from their environment). Agents can then be read as minimizing a variational free energy functional of (approximate Bayesian) beliefs over unobservable causes or states Q(s), given by
(1)Q(sτ,uτ)=argminQF(2)F=EQ[lnQ(sτ,uτ)︸posterior−lnP(oτ|sτ,uτ)︸likelihood−lnP(sτ,uτ)︸prior](3)=DKL[Q(sτ,uτ)||P(sτ,uτ|oτ)]︸divergence−lnP(oτ|m)︸log evidence(4)=DKL[Q(sτ,uτ)||P(sτ,uτ)]︸complexity−EQ[lnP(oτ|sτ,uτ)]︸accuracy

Agents interact with the environment, updating their beliefs to minimize variational free energy. The variational free energy provides an upper bound on the log evidence for the generative model (also known as marginal likelihood), which can be understood in terms of optimizing Bayesian beliefs to provide a simple but accurate account of the sensorium (i.e., minimizing complexity while maximizing accuracy). This is sometimes referred to as self-evidencing [25].

Priors over action are based on the free energy expected following an action, such that the most likely action an agent commits to can be expressed as a softmax function of expected free energy:(5)P(u)=σ(−γ·G(u))(6)G(u)=EQu[lnQ(sτ+1|u)−lnQ(sτ+1|oτ+1,u)−lnP(oτ+1|c)](7)=EQu[lnQ(sτ+1|oτ+1,u)−lnQ(sτ+1|u)]︸expected information gain−EQu[lnP(oτ+1|c)]︸expected value(8)=DKL[Q(oτ+1|u)||P(oτ+1|c)]︸risk−EQu[lnQ(oτ+1|sτ+1,u)]︸ambiguity

Here, G(π) is the expected free energy for policy u and γ is a precision parameter influencing the stochasticity of action selection. Effectively, actions are selected based on their expected value, which is the expected log likelihood of preferred observations, and their epistemic value, which represents the expected information gain. The expected free energy can also be rearranged in terms of risk and ambiguity, namely, the divergence between anticipated and preferred outcomes and the imprecision of outcomes, given their causes. A comparison of Equations (2) and (3) shows that risk is analogous to expected complexity, while ambiguity can be associated with expected inaccuracy. In summary, agents engage with their world by updating beliefs about the hidden or latent states, causing observations while, at the same time, acting to solicit observations that minimize expected free energy, namely, minimizing risk and ambiguity.

It is important to note that the active inference framework is not a metaphysical approach or framework but rather a physics modeling framework. It only assumes the rest of contemporary physics as a background (classical, statistical, and quantum mechanics) and answers the question: given this background, what does it mean for something to be reliably re-identifiable by an external observer as the thing that it is? It makes no metaphysical claims per se (see [26,27,28]).

## 4. Active Inference and Time Consciousness

We have previously [1] framed Husserl’s descriptions of time consciousness in terms of (Bayesian) belief updating, while further work proposed a mathematical reconstruction of the core notions of Husserl’s phenomenology of time consciousness—retention, primal impression, protention, and the constitution of disclosure of objects in the flow of time consciousness—using active inference [2]. See also [29] for related work. Active inference foregrounds a manner in which previous experience updates an agent’s (Bayesian) beliefs and thereby underwrites behaviors and expectations, leading to a better understanding of the world.

Descriptions of temporal thickness from Husserl’s phenomenology are highly compatible with generative modeling in active inference agents. To connect the phenomenological ideas reviewed above with active inference, consider again how active inference agents update their beliefs about the world based on the temporal flow structure. In active inference, agents continuously update their posterior beliefs by integrating new observations with their existing beliefs. This belief updating involves striking a balance between maintaining the agent’s current beliefs and learning from new information. Retention and primal impressions relate to the fact that, in active inference, all past knowledge contributes to shaping beliefs about the present state of the world. In short, active inference effectively bridges prior experiences with current expectations. In this setting, retentions are formalized as the encoding of new information about the world under a generative model. Primal impressions are formalized as the new data that agents sample over time, forcing an agent to update their beliefs about the world, leading to a better understanding of the environment and allowing for more effective information and preference-seeking behavior. Thus, the consciousness of inner time can be modeled as active inference, where prior beliefs (sedimented retentions) are intermingled with ongoing sensory information (primal impression) and contextualized by an unfolding, implicit, and future-oriented anticipation of what will be sensed next (protention).

### 4.1. Mapping Husserlian Phenomenology to Active Inference Models

As illustrated in Table 1 and previously explored in [1], active inference comprehensively maps to Husserlian phenomenology: observations represent the hyletic data, namely, sensory information, that set perceptual boundaries but are not directly perceived. The hidden states correspond to the perceptual experiences, namely, the data that are inferred from the sensory input. The likelihood and prior transition matrices are associated with the idea of sedimented knowledge. These matrices represent the background understanding and expectations that scaffold perceptual encounters. The preference matrix evinces a similarity to Husserl’s notions of fulfillment or frustration. It represents the expected results or preferred observations. The initial distributions represent the prior beliefs of the agent that are shaped by previous experiences. In particular, the habit matrix resonates with Husserlian notions of horizon and trail set. Collectively, these aspects of the generative model entail prior expectations and the possible course of action.

### 4.2. Intersubjectivity and Intentionality

There is a body of related work on shared intentionality, intersubjectivity, and joint/shared attention that is relevant in this context (see, e.g., [30,31,32,33]). Joint attention is defined as the state in which two or more individuals focus on the same object or event. It involves mutual understanding and the coordination of attention, underpinned by cognitive processes that enable individuals to recognize and follow each other’s gaze or attentional focus, thus establishing a shared point of interest. Shared intentionality is the capacity to share psychological states, including intentions, beliefs, and goals, with others in order to coordinate.

These concepts have been developed in a Bayesian framework, which is quite germane to our own approach. Intersubjectivity here is seen as a consequence of agents conforming to the dependence structures harnessed in the same (or similar) generative models and of the capacity of the agents to consider events and their sequentiality in a similar way and coordinate around them. For instance, ref. [34] has argued that shared intentionality depends on agents having a shared representation of the joint goal and its context. This is a key aspect of intersubjectivity, as the agents maintain beliefs about a shared goal that they are trying to accomplish together. Agents continuously update their beliefs about the joint goal, which act as a context based on observing each other’s actions. This involves inferring the other’s intentions and goals based on their movements. Through this kind of interactive inference, the agents align their representations and behaviors over time. This models the intersubjective alignment and “coupling” of representations that enable successful joint action in humans.

In a different study, ref. [35] investigated ant colony foraging behavior using an active inference model. The study modeled ant behavior in a T-maze paradigm, simulating how ants discover food sources and communicate these locations to the colony through pheromone trails. This behavior is formalized via active inference, emphasizing the ants’ ability to make predictions and inferences about their environment based on sensory inputs and prior knowledge. This study also illustrates how shared representations and goals (e.g., locating food sources) facilitate coordinated actions within the ant colony. The ants’ behavior, driven by hierarchical Bayesian inference, showcases a form of temporal integration where actions are predictive of the future, based on the anticipated positions of resources and the inferred intentions of other ants.

We draw partly from this work to justify our claim that the structure of temporality enables coordination, but we take a significant step beyond merely sharing similar predictive models. We suggest that agents do not just align their predictions of the future, but rather, also integrate the predictive models of others into their own. This integration reflects a deeper level of intersubjectivity, where the contents that are protended include anticipatory representations of other agents’ future states of mind. Such integration is akin to a form of theory of mind, where understanding and predicting the mental states of others—including their intentions and future actions—are essential for complex social interactions. It also entails that the distinction between self and other gets blurrier as we move up a predictive hierarchy, explaining the co-constitutive nature of the self.

### 4.3. An Active Inference Approach to Shared Protentions

Active inference often involves agents making inferences about each other’s mental states by attributing cues to underlying causes [36]. The emergence of communication and language in collective or federated inference provides a concrete example of how these cues become standardized across agents [37]. Individual agents leverage their beliefs to discern patterns of behavior- and belief-updating in others. This entails a state of mutual predictability that can be seen as a communal or group-level reduction in (joint) free energy [36,38]. When agents are predictable to each other, they can anticipate each other’s actions in a complementary fashion, in a way that manifests as generalized synchrony [39]. This is similar to how language emerges in federated inference as a tool for minimizing free energy across agents in a shared econiche [40].

As agents in a group exhibit similar behaviors, they generate observable cues in the environment (e.g., an elephant path through a park) that guide other agents towards the same generative models and nudge agents towards the same behavior. This has been discussed in terms of “deontic value”, which scores the degree to which an agent’s observation of a behavior will cause that agent to engage in that behavior [40]. Alignment can thus be achieved by agents that share similar enough goals and exist in similar enough environments, thereby reinforcing patterns of behavior and epistemic foraging for new information [36,40]. Individual agents perceive the world, link observable “deontic cues” to the latent states and policies that cause them, and observe others, using these cues to engage in situationally appropriate ways with the world. These deontic cues might range from basic road markings to intricate semiotics or symbols such as language [41].

Consider, for example, someone wearing a white lab coat. You and the person wearing the lab coat share some similarities that lead you to believe that the coat means the same thing to them as it does to you. You know that lab coats are generally worn for scientific or medical purposes (sedimented retentions scaffolded by the cognitive niche). The person wearing the lab coat is standing in a street near a hospital building. From all these cues, without ever wearing a lab coat yourself or being a doctor, you can make a pretty good guess that this individual is a doctor.

Similarly, gathering *interoceptive* cues to infer one’s own internal states can enable individuals to make sense of other people’s behavior and allow them to infer the internal states of others [42,43]. Within a multi-agent system, the environment is complex, encompassing both abiotic and social niches. The abiotic niche is the physical and inanimate components of the environment, whereas the social niche comprises the interactions and observations resulting from the activities of other agents. This differentiation underscores that the environment is not exclusively shaped by agents but rather is shaped by an intricate, self-sustaining interaction between living and non-living things. This suggests that agents not only acquire knowledge about their environment to efficiently understand and navigate it, but they also acquire knowledge about others and, implicitly, themselves in tandem. These ideas provide some framing for the core question that will concern us presently: how should we model such shared protentions? To model shared protentions effectively, the use of category theory becomes a key epistemological resource. The mathematical precision and structural complexity of category theory provide a sophisticated framework for comprehending the cohesive behaviors of agents with regard to shared objectives or future perspectives. It explores fundamental relationships among things like the characteristics of agents’ interactions, goals, protentions, and the organization of their environmental resources. This theoretical framework allows for a formal and scalable understanding of shared protentions, highlighting the interconnections and relational dynamics between individuals in a complex setting.

## 5. Category Theoretic Description of Shared Protentions in Active Inference Ensembles

We have reviewed active inference-based approaches to time consciousness and, in particular, the construct of a shared protention. Here, we propose a category theoretic formulation of shared protentions among an ensemble of active inference agents [44] combining two main ideas: first, a notion of agent derived from categorical systems theory [45], whose boundary (or Markov blanket) is described using *polynomial functors*; second, the concept of *sheaf*, to account for agents with shared beliefs that may thus be “glued together”.

We do not pretend to give a detailed mathematical exposition of either of these concepts here and instead refer the interested reader to [46] on polynomial interaction and [47] on the basic ideas of sheaf theory. For our purposes, it will be sufficient to know some basic concepts from set theory (the notions of *disjoint union* and *intersection*) and the basic definitions of *category* and *functor*, which we now review.

Categories capture the mathematical essence of composition, the process by which many parts make a whole. A *category* X is thus determined by a collection of objects, denoted $X_{0}$, and, for each pair (a,b) of objects, a set X(a,b) of *morphisms* from *a* to *b*. We denote such a morphism by f:a→b, and say it has *source a* and *target b*. Morphisms with compatible source and target may be composed, so that f:a→b and g:b→c yield g∘f:a→c, and each object *a* is assigned an *identity* morphism, $id_{a}:a\to a$. The morphisms of a category are required to satisfy two axioms: *unitality*, saying $f\circ id_{a}=f=id_{b}\circ f$; and *associativity*
h∘(g∘f)=(h∘g)∘f, meaning we can simply write h∘g∘f for consecutive composition. A basic example of a category is the category Set, whose objects are sets *X* and whose morphisms f:X→Y are functions f(x)=y.

A *functor* is a morphism between categories. If we think of a category as like “a set where there may be relationships (morphisms) between points”, then a functor is like a function that preserves the structure of those relationships. Formally, if C and D are categories, then a functor F:C→D is a mapping $F_{0}:C_{0}\to D_{0}$ along with a family of functions $F_{a,b}:C\left(a,b\right)\to D\left(F_{0}a,F_{0}b\right)$, indexed by the objects a,b of C; one typically drops the subscripts and infers them from the context. These mappings must satisfy the axioms of *functoriality*: F(g∘f)=F(g)∘F(f) and $F\left(id_{a}\right)=id_{Fa}$ for all morphisms f,g and objects *a* in C. Each object *a* in a category C induces a functor C(a,−):C→Set, which maps each object *b* to the set C(a,b) of morphisms a→b, and which sends each morphism g:b→c to the function g∘(−):C(a,b)→C(a,c), which acts by post-composition, f↦g∘f. Beyond these “representable” functors, both polynomial functors and sheaves are also special kinds of functor.

### 5.1. Polynomial Functors

At school, we learn about polynomial functions, such as $f\left(x\right)=x^{2}+3x+2$; a polynomial *functor* is to this concept precisely what a functor is to a function. Formally, one merely changes the variables, coefficients, and exponents in the expression from numbers to sets (this replacement may be seen to generalize polynomial functions if we note that a number such as 3 may be seen to stand for a set {∗,∗,∗} of the same cardinality). In an expression such as $y^{A}+By+C$, we interpret the exponential $y^{A}$ as the representable functor $X\mapsto X^{A}:=\mathrm{Set}\left(A,X\right)$, By as the product functor X↦B×X and + as the disjoint union of sets, so that altogether, the expression encodes the functor $X\mapsto X^{A}+B\times X+C$.

Every polynomial can be written in the form of a sum (disjoint union) of representable functors, $\sum_{i:I}y^{p[i]}$, for some indexing set *I* and collection of exponents {p[i]}, for example, we can write By as $\sum_{b:B}y^{1}$, where 1 is the 1-element set {∗}. Therefore, we will henceforth summarize the data of a polynomial *p* as
$$p=\sum_{i:p(1)}y^{p[i]}$$
where we now write p(1) for the indexing set.

The mathematics of polynomial functors supplies a perhaps surprisingly rich formalism for describing interacting systems such as intelligent agents. We can think of a polynomial *p* as describing the “interface” or “boundary” of such a system: each element *i* of p(1) represents a possible shape or configuration that the system may adopt or the possible actions that it may take, and each exponent p[i] represents the set of possible “inputs” that it may expect (such as sense data), having adopted configuration *i*.

Because the type of expected sense data may depend on the configuration adopted (just as you do not expect to “see” when you close your eyes), this generalizes the usual notion of a *Markov blanket* in active inference to something more dynamic. We can thus model an active inference agent with boundary polynomial *p* as predicting the activity of its boundary *p*. That is to say, we collect the exponent sets p[i] together into their disjoint union $\Sigma p:=\sum_{i:p(1)}p\left[i\right]$ and then understand the agent as predicting a distribution over the whole set Σp. This amounts to predicting both its configurations i:p(1) (hence, its actions) and, compatibly, its sense data in each p[i]. If we restrict each p[i] to being the same (so there is no dependence of sense data on configuration), then we can recover the standard Markov blanket: if we set p[i]=S to be the sense data and p(1)=A the actions, then $\sum_{a:A}S=A\times S$.

Being a category theoretic formalism, one does not just have objects (polynomials) but also morphisms between them. These encode the data of how agents with polynomial interfaces may interact, in particular, they encode how systems may be “nested” within each other. Thus, a morphism φ:p→q encodes how a system with boundary *p* may be nested within a system with boundary *q*, and consists of a pair $\left(\varphi_{1},\varphi^{♯}\right)$ of “forwards” function $\varphi_{1}:p\left(1\right)\to q\left(1\right)$ (that encodes how *p*-configurations or *p*-actions are translated into *q*-configurations) and a family of “backwards” functions $\varphi_{i}^{♯}:q\left[\varphi_{1}\left(i\right)\right]\to p\left[i\right]$ (that encodes dually how *q*-sense data are translated into *p*-sense data). Polynomials and their morphisms collect into a category: Poly.

Now, a morphism p→q represents simply nesting a *p*-system within a *q*-system; but often, as here, we wish to consider how multiple agents form a coherent collective, which means we need a way to encode multiple agents’ polynomials as a single polynomial. For this, we can use the *tensor* of polynomial functors, $p⊗p^{\prime }$, which places the two interfaces *p* and $p^{\prime }$ “side by side”. Formally, we define $p⊗p^{\prime }$ as the polynomial $\sum_{i:p(1)}\sum_{i^{\prime }:p^{\prime }\left(1\right)}y^{p\left[i\right]\times p^{\prime }\left[i^{\prime }\right]}$. With this definition, we can understand a morphism $p⊗p^{\prime }\to q$ as representing how systems *p* and $p^{\prime }$ come together to form a system with boundary *q*.

This is not yet enough for our purposes; we also wish to model systems that recursively predict the beliefs of other agents in their environment. Behaviorally, this means predicting how the other agents are going to act given their perceptions, which in turn means predicting the patterns of interaction within the environment. And, formally, this means “internalizing” these patterns into a single polynomial.

Thus, given polynomials *p* and *q*, we can define the corresponding *hom* polynomial
$$\left[p,q\right]:=\sum_{\varphi :\mathrm{Poly}(p.q)}y^{\sum_{i:p(1)}q[\varphi_{1}\left(i\right)]}$$The set of configurations of [p,q] is the set of morphisms p→q, so to adopt a [p,q]-configuration is to adopt a particular pattern of interaction.

Dually, the “sense-data” associated with a particular pattern of interaction φ is given by the configurations of the “inner” system *p* and, for each such configuration i:p(1), the corresponding sense data $q[\varphi_{1}\left(i\right)]$ for the outer system *q* in the configuration implied by *i* via φ.

A prediction over [p,q] is thus a prediction over the set
$$\Sigma \left[p,q\right]=\sum_{\varphi :\mathrm{Poly}(p.q)}\sum_{i:p(1)}q[\varphi_{1}\left(i\right)]$$
that is, a distribution over patterns of interaction Poly(p,q), inner configurations p(1), and outer sense data $\sum_{i}q\left[\varphi \left(i\right)\right]$. By way of an example, if we assume that the outer system *q* is “closed” (with no further external environment), then that is to say that it has the trivial interface q=y with only one configuration (“being”) and no non-trivial sense data. A morphism p→y corresponds to a function $p\left(1\right)\to \sum_{i}p\left[i\right]$ (strictly speaking, a *section* of the bundle $\sum_{ip(1)}p\left[i\right]\to p\left(1\right)$), which encodes how the environment responds with sense data, given the *p*-system’s actions. Thus, a prediction over [p,y] is a prediction of the environment’s response, along with a prediction of “how to act”.

### 5.2. Generative Models via Polynomials with Stochastic Feedback

In order to extend the standard formalism of active inference to this more general setting, we need to add two more ingredients to the Poly mix: stochasticity and dynamics. And to keep the presentation as simple as possible, we will focus on discrete models (discrete in time and space). The key component of a generative model in discrete active inference is a partially observable Markov decision process (POMDP), which encodes an agent’s beliefs about the (stochastic) dynamics of its environment and the observations (sense data) that these generate. POMDPs can be expressed very succinctly in the stochastic Poly setting.

To incorporate stochasticity into Poly, one can adjust the morphisms of polynomials so that they have stochastic backward components: *stochastic feedback*. Thus, in the resulting category StochPoly, a morphism φ:p→q consists of a forward function $\varphi_{1}:p\left(1\right)\to q\left(1\right)$ along with a p(1)-indexed family of stochastic backward maps $\varphi_{i}^{♯}:q\left[\varphi_{1}\left(i\right)\right]⇝p\left[i\right]$. In the discrete case, a stochastic map c:X⇝Y is equivalent to a function c:X→DY, where DY is the set of (finitely supported) probability distributions on *Y*; equivalently, the set of conditional probability distributions c(y|x). Thus, $\varphi^{♯}$ is given by a p(1)-indexed family of conditional probability distributions.

A discrete-time dynamical system with state space *S* and polynomial interface (“Markov blanket”) *p* is a polynomial morphism $\vartheta :Sy^{S}\to p$. This consists of an “output” function $\vartheta_{o}:S\to p\left(1\right)$, which picks a configuration of the Markov blanket for every state in *S*, and a family of “update” functions $\vartheta_{s}^{u}:p\left[\vartheta_{o}\left(s\right)\right]\to S$, which update the state given inputs corresponding to the current state.

Instantiating this recipe in StochPoly and letting *p* be a “monomial”, such as $Oy^{U}$, we find that a system $\rho :Sy^{S}\to Oy^{U}$ corresponds to a POMDP with state space *S*, observation space *O*, and action space *U*. That is, it is given by an output function $\rho_{o}:S\to O$ (which encodes how the agent believes its observations are generated) and a transition matrix $\rho^{u}:S\times U⇝S$, which may be written $\rho^{u}\left(s_{\tau +1}|s_{\tau },u\right)$.

Classical active inference then couples this generative model $\rho :Sy^{S}\to Oy^{U}$, which encodes the environment’s dynamics, with a “control” system α, which encodes the agent’s interaction with the environment, i.e., how actions are generated. Classically, the coupled agent–environment system is then assumed to be closed, that is, it is assumed to have the trivial polynomial interface *y*. This means that, in general, the agent’s control system α is of the form $S^{*}y^{S^{*}}\to \left[p,y\right]$, since there is a canonical morphism [p,y]⊗p→y. This system α, along with the priors necessary to initialize the systems, is typically constituted by the remaining data in Table 1.

### 5.3. Beyond Classical Active Inference: Multi-Agent Systems

Having sketched how active inference may be expressed using stochastic polynomial dynamics, we see that there are natural loci for generalization. A first is that an agent (or its environment) need not have a monomial interface; we can now move beyond the classical POMDP framework. One way to make use of this is to describe agents who believe themselves to be situated among other agents: a key step towards a mathematical description of shared protentions. In such a context, a single agent need not believe simply that they and their (amorphous) environment constitute the whole “closed” universe, represented by the polynomial *y*. Rather, an agent who believes themselves among a group may believe that the universe is only closed when accounting for the behavior of the whole group.

Thus, letting the agents’ Markov blankets be denoted by $p_{j}$, we may replace the trivial polynomial *y* with the hom $\left[{⊗}_{j}\;p_{j},y\right]$, so that a single agent’s control system $\alpha_{i}$ obtains the type $S_{i}^{*}y^{S_{i}^{*}}\to \left[p_{i},\left[{⊗}_{j\neq i}\;p_{j},y\right]\right]$. It is possible to prove an isomorphism of polynomials [p,[q,r]]≅[p⊗q,r], and so $\alpha_{i}$ equivalently has the interface $\left[{⊗}_{j}\;p_{j},y\right]$, and this type signature holds for all agents in the group. Then, for consistency, each agent’s generative model $\gamma_{i}$ must have the type $S_{i}y^{S_{i}}\to {⊗}_{j}\;p_{j}$, which means that each agent believes that the dynamics of the hidden states depend on the actions of all the agents and, likewise, that the hidden states determine the observations of all the agents.

Such agents thus predict not only their individual actions but those of their companion agents, along with how the environment will respond to all of them. The foregoing analysis can be repeated to arbitrary levels of nesting so that it constitutes a starting point for a formal “theory of mind” and, indeed, a starting point for an account of agents that model each other’s protentions.

### 5.4. A Sheaf-Theoretic Approach to Multi-Agent Systems

In the preceding two sections, we described how an ensemble of agents may predict each other’s behavior by instantiating a family of polynomial generative models. However, there is nothing in that formalism that pushes the agents’ beliefs to be in any way compatible; they need not *share* protentions. Indeed, a true *collective* of agents should be a group of agents that have “overlapping” world models that are sufficiently cohesive to promote the development of common intentions among individuals.

In order to describe agents with such shared beliefs, we propose upgrading the formalism using the mathematical tools of sheaf and topos theory. Sheaves are, in some sense, the canonical structure for distributed data [47], and tools from sheaf theory allow us to describe agents that communicate in order to reach a consensus [48].

In more detail, a sheaf over a topological space constitutes a systematic method of keeping track of how “local” data or qualities, defined on open subsets, can be reliably concatenated to represent a “global” situation. This attribute renders them highly valuable in comprehending the varied and potentially contradictory convictions, perspectives, and forecasts of individual agents within a multi-agent system. Sheaves enable the analysis of both the diversity and agreement among various agents’ perspectives on the environment, and their ability to alter over time is crucial for adjusting and reacting to system modifications. Sheaves formalize the concept of “shared experience” among agents, which is essential for reaching a consensus on the structure of the external world.

Mathematically, a sheaf *F* is an assignment of data sets to a space *X*, such that the assignment “agrees on overlaps”, meaning that, if we consider overlapping subsets *U* and *V* of *X*, then F(U) and F(V) agree on the overlap U∩V. A little more formally, if we consider there to be a morphism $U\to U^{\prime }$ whenever $U^{\prime }\subseteq U$, we obtain a category O(X) whose objects are (open) subsets of *X* and whose morphisms are such (“opposite”) inclusions. Then, a sheaf *F* is a functor O(X)→Set, such that, whenever *U* and *V* cover *W* (as when W=U∩V) so that there are morphisms $\iota_{U}:U\to W$ and $\iota_{V}:V\to W$ in O(X), then, if u∈U and v∈V, there is a unique w∈W such that $F\left(\iota_{U}\right)\left(u\right)=w=F\left(\iota_{V}\right)\left(v\right)$. The category of sheaves on *X* forms a subcategory Sh(X) of the category of functors O(X)→Set.

Now, a space such as *X* is itself an object of a *category of spaces*
Spc, whose morphisms are the appropriate kind of functions between spaces (e.g., continuous functions between topological spaces), and when Spc has enough structure (such as when it is a *topos*), there is an equivalence between Sh(X) and the category Spc/X of *bundles* over *X*, whose objects are morphisms $\pi_{E}:E\to X$ in Spc and whose morphisms $f:\pi_{E}\to \pi_{F}$ are functions f:E→F, such that $\pi_{E}=\pi_{F}\circ f$. We can use this equivalence to lift the models of the previous section to the world of sheaves, as we now sketch (to see one direction of the equivalence, observe that, given a bundle $\pi_{E}:E\to X$, we can obtain a sheaf by defining F(U) to be the pullback of $\pi_{E}$ along the inclusion U↪X).

A bundle $\pi_{E}:E\to X$ thus may itself be seen as representing a type (or collection) of data that vary over the space *X*; for each x∈X, there is a *fibre*
$E_{x}$ encoding the data relevant to *x*. In this way, each polynomial $\Sigma p=\sum_{i:p(1)}p\left[i\right]$ yields a “discrete” bundle $\sum_{i:p(1)}p\left[i\right]\to p\left(1\right)$ that maps (i,x) to *i*. But the polynomials of the preceding sections are in no way related to another ambient spatial structure. For instance, one might expect that the internal state space *S* of an agent’s generative model is structured as a model of the agent’s external environment, which is likely spatial. Likewise, the type of available configurations p(1) may itself depend on where in the environment the agent finds themselves (consider that we might suppose this *S* to also encode task-relevant information).

This suggests that the model’s configuration space p(1) should itself be bundled over *S*, so that the polynomial *p* takes the form $\sum_{i}p\left[i\right]\to p\left(1\right)\to S$ as a bundle, or, better put, as an object in Spc/S. In order for this to make sense, we need to be able to instantiate the category Poly in Spc/S rather than Set. This is possible if Spc has enough structure (it must be *locally Cartesian closed*, which it will be if it is a topos), as we have assumed. Then, we can define a *spatial generative model* on the interface *p* over *X* to be a (stochastic) system $Sy^{S}\to p$ as before, but now instantiated in Spc/S. Explicitly, this is a pair of maps $\left(\gamma_{o},\gamma^{u}\right)$ making the following diagram commute (where deterministic maps compose after stochastic ones by pushforward):

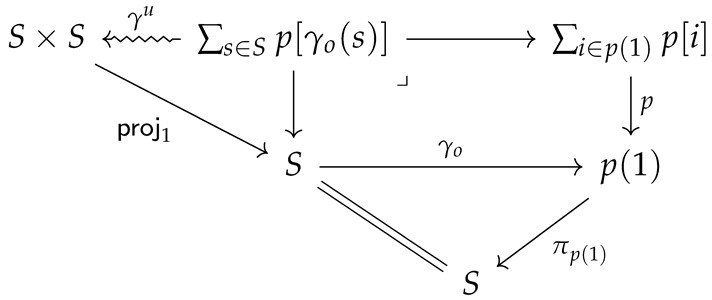

The commutativity of this diagram ensures, in particular, that the spatial structure of *S* is respected. For example, if p(1) represents the observation space of the agent, then observations may only be predicted “where they may be made”.

Now, in this spatially enhanced setting, we may recapitulate the polynomial theory-of-mind of the preceding section and suppose that each agent *j* is equipped with a spatial generative model of the form $S_{j}y^{S_{j}}\to {⊗}_{i}p_{i}$. If we additionally suppose that the collection of agents’ model spaces $\left\{S_{j}\right\}$ covers a (perhaps larger) space *X*, then we can in turn ask whether it is possible to glue these models $S_{j}\to {⊗}_{i}p_{i}$ together accordingly to form a “sheaf of world models” *W*.

If it *is* possible, then we may say that the agents inhabit a shared universe, and thus, with appropriate generative models, may be said to *share protentions*. Conversely, if it is *not* possible, then we may ask, what is the obstruction? In this case, there must be some *disagreement* between the agents. But sheaf theory supplies tools for overcoming such disagreements [49,50], and thus to communicate to reach a consensus [48]. Even if the disagreements are fundamental, it is usually possible to derive dynamics that will yield as close to a sheaf as possible [51]. In future work, we hope to apply these methods to multi-agent, active inference model in order to demonstrate this consensus-building.

#### A Note on Toposes

Sheaves collect into categories called *toposes*. A topos is a category that has both a spatial and logical structure [52], allowing for the expression of logical propositions and deductions within it. Topos theory, extending beyond sheaf theory, provides a more holistic and abstract framework. Each topos is like a “categorified space” and comes with an *internal logic* and language whose expressions are relative to the space that the topos models, thereby enabling a more profound exploration of the conceptual structures within these spaces. In this way, each topos can be thought of as a “universe”, where the truth of propositions may depend on where they are uttered. For example, the topos Spc/X assumed above represents “the universe of the space *X*”, known as the “little topos” or “petit topos” of *X*.

The tools of sheaf theory naturally extend to toposes. Thus, in the foregoing discussion, we considered a collection of agents with internal world models $\left\{X_{j}\right\}$, which, in turn, induce toposes $\mathrm{Spc}/X_{j}$ that we may consider gluing into a “shared universe” or “consensus topos” Spc/W according to their topology or interaction pattern. Perhaps, in the end, we may consider this shared universe to be the agents’ understanding of their actual universe, socially constructed.

## 6. Closing Remarks

Our paper introduces the integration of Husserlian phenomenology, active inference in theoretical biology, and category theory. We have formalized collective action and shared goals using mathematical tools of increasing generalization. We were able to anchor this formalism in phenomenology by delving into Husserl’s phenomenology of inner time-consciousness, emphasizing retention, primal impression, and protention. We then proposed that these concepts could be connected in the formation of shared goals in social groups. With a short overview of active inference, we cast the action–perception loop of cognitive agents as variational inference, furnishing an isomorphic construct to time consciousness. Building on this introduction, we were able to review the relationship between Husserl’s time consciousness and active inference established in a previous paper, showing how past experiences and expectations influence present behavior and understanding. We then proceeded to leverage a category theory to model shared protentions among active inference agents, using concepts like polynomial functors and sheaves. These sophisticated tools were necessary to account for the complexity of shared protentions, leveraging existing tools of category theory. Our paper achieves a conceptual and mathematical image of the interconnection among agents, enabling them to coordinate in large groups across spatiotemporal scales. It dissolves the boundaries between externalist and internalist perspectives by demonstrating the intrinsic connections of perceptions extended in time. This formalization elucidates how agents co-construct their world and interconnect through this process, offering a novel approach to understanding collective action and shared goals. As is the case with most work in computational neuroscience and psychology, this paper is only one step in a broader, multi-step, iterative process where we transition from abstract theory- and model-building (as in this paper) to experimental validation. The proposal in this paper represents a point of departure for future work on the phenomenology of shared intentionality, indeed, it is formulated quite broadly and does not provide a specific testable model yet. Moving towards such a specific model making testable hypotheses will be left for future follow-up work.

## Figures and Tables

**Table 1 entropy-26-00303-t001:** Parameters used in the general model under the active inference framework and their phenomenological mapping.

Parameter	Description	Phenomenological Mapping
o∈O	Observations that capture the sensory information received by the agent	Represents the hyletic data, setting perceptual boundaries but not directly perceived
s∈S	Hidden states that capture the causes for the sensory information: the latent or worldly states	Corresponds to perceptual experiences inferred from sensory input
P(o|s)=Cat(A)	Likelihood matrix that captures the mapping of observations to (sensory) states	Associated with sedimented knowledge, representing background understanding and expectations
$P\left(s|s_{-1},\mu \right)=Cat\left(B\right)$	Transition matrix that captures the mapping for how states are likely to evolve	Linked to sedimented knowledge, shaping perceptual encounters
$P\left(o_{+1}|c\right)=Cat\left(C\right)$	Preference matrix that captures the preferred observations for the agent, which drive their actions	Similar to Husserl’s notions of fulfillment or frustration, representing expected results or preferences
$P\left(s_{0}\right)=Cat\left(D\right)$	Initial distribution that captures the priors over the hidden states	Represents prior beliefs shaped by previous experiences and current expectations
P(μ)=Cat(E)	Habit matrix that captures the prior expectations for initial actions	Connected to Husserlian notions of horizon and trail set, symbolizing prior expectations
π	Policy matrix that captures the potential policies that guide the agent’s actions, driving the evolution of the B matrix	Symbolizes the possible course of action, influenced by background information and values

## Data Availability

No new data were created or analyzed in this study. Data sharing is not applicable to this article.

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
