# Peer review of "Shared Protentions in Multi-Agent Active Inference"

_entropy, 2024, doi:10.3390/e26040303_

Round 1
Reviewer 1 Report
Comments and Suggestions for Authors
This is an interesting article, well written and insightful, another variation on the broad applicability of active inference. This time addressing philosophical considerations about social interaction and shared understanding. There are lots of assumptions and a sophisticated mathematical framework (which I cannot fully assess) but no empirical investigation or evidence whatsoever. As an academic exercise this paper is interesting and worthwhile, but also a little bit repetitive with all the many other papers on active inference that are mainly mathematical/theoretical.
I have only a comment: While the AIF framework is beautiful, I am a fan of it, I wonder (actually doubt) whether this categorical incarnation of it can actually be applied to such entangled domains as real social interaction. It may be that there are no categories 'inside' an agent. It could be that categories only emerge on the output of an agent, in their interaction (outside the MB), and are results of our 'academic' view on it. Categories and compositionality may be a good tool for us to understand, but the processes themselves might be actually quite different. It could be that there is a superposition of lots of possibilities inside the agent which collapse only at the output into what we consider to be categories. Millière & Buckner (https://arxiv.org/pdf/2401.03910.pdf) talk about continuous compositionality. They look into LLMs of actually working models. Do you see a possibility to reconcile those views/ideas?
The authors say "In future work, we hope to apply these methods to multi-agent active inference model, in order to demonstrate this consensus-building". It is not clear what this 'demonstration' would be, I hope that the authors take real data, a real implementation and provide a 'real' evaluate their model.
in line 173, it was unclear to me what "Here, G(π)" is.
Author Response
This is an interesting article, well written and insightful, another variation on the broad applicability of active inference.
We thank the reviewer for their kind words and useful comments, which have helped us to improve the paper.
This time addressing philosophical considerations about social interaction and shared understanding. There are lots of assumptions and a sophisticated mathematical framework (which I cannot fully assess) but no empirical investigation or evidence whatsoever. As an academic exercise this paper is interesting and worthwhile, but also a little bit repetitive with all the many other papers on active inference that are mainly mathematical/theoretical.
We largely agree with this assessment and appreciate the reviewer raising this point. We have decided to better justify the paper. As is the case with most work in computational neuroscience and psychology, this paper is only one step in a broader, multi-step and iterative process where we transition from abstract theory- and model-building (as in this paper) to experimental validation. The proposal in this paper represents a point of departure for future work on the phenomenology of shared intentionality; indeed, it is formulated quite broadly, and doesn’t provide a specific testable model yet. Moving towards such a specific model making testable hypotheses will be left for future follow up work. Speaking also to the point raised by the reviewer below, we aim to follow up this work with experimental work, to validate the theoretical model that we propose here.
We have added:
“As is the case with most work in computational neuroscience and psychology, this paper is only one step in a broader, multi-step and iterative process where we transition from abstract theory- and model-building (as in this paper) to experimental validation. The proposal in this paper represents a point of departure for future work on the phenomenology of shared intentionality; indeed, it is formulated quite broadly, and doesn’t provide a specific testable model yet. Moving towards such a specific model making testable hypotheses will be left for future follow up work.”
I have only a comment: While the AIF framework is beautiful, I am a fan of it, I wonder (actually doubt) whether this categorical incarnation of it can actually be applied to such entangled domains as real social interaction. It may be that there are no categories 'inside' an agent. It could be that categories only emerge on the output of an agent, in their interaction (outside the MB), and are results of our 'academic' view on it. Categories and compositionality may be a good tool for us to understand, but the processes themselves might be actually quite different. It could be that there is a superposition of lots of possibilities inside the agent which collapse only at the output into what we consider to be categories. Millière & Buckner (https://arxiv.org/pdf/2401.03910.pdf) talk about continuous compositionality. They look into LLMs of actually working models. Do you see a possibility to reconcile those views/ideas?
The authors say "In future work, we hope to apply these methods to multi-agent active inference model, in order to demonstrate this consensus-building". It is not clear what this 'demonstration' would be, I hope that the authors take real data, a real implementation and provide a 'real' evaluate their model.
While we agree with the general spirit of the reviewer’s remarks, we find their comments on categories a bit confusing. In category theory, a category is just a collection of objects and their relations (i.e., maps between objects), along with composition operations that are governed by associativity and identity rules. Category theory provides us with a way of expressing generic mathematical structure, in a way that enables us to identify common structures in formalisms. In this technical sense, category theory is absolutely general and applies to every area of mathematical inquiry, revealing the underlying structures of the formalisms being studied. The reason we mention this is that, generally speaking, if agents can be described as having or conforming to any mathematical structure at all, then that structure can be studied through the lens of compositionality using category theory. The question of the locus of these properties—whether these categories describe properties that are best conceived of intrinsically as being internal to an agent, or rather as structures describing the interactions between an agent and an observer, or as an expression of the relational properties between agent and environment as they appear to an external observer—is thus arguably orthogonal to the question whether category theory can be applied to make sense of these structures. In every case, a priori, it can.
That said, we agree with the reviewer that “Categories and compositionality may be a good tool for us to understand, but the processes themselves might be actually quite different.” However, we see no substantial disagreement between the reviewer’s position and ours here, and hope we have not indicated anything that would suggest otherwise. Indeed, the active inference framework is not a metaphysical approach or framework, but rather a physics modeling framework. It only assumes the rest of contemporary physics as a background (classical, statistical, and quantum mechanics) and answers the question: given this background, what does it mean for something to be reliably re-identifiable by an external observer, as the thing that it is—it makes no metaphysical claims per se. See Ramstead, Sakthivadivel, & Friston (2022); Ramstead et al., (2023).
We have added:
It is important to note that the active inference framework is not a metaphysical approach or framework, but rather a physics modeling framework. It only assumes the rest of contemporary physics as a background (classical, statistical, and quantum mechanics) and answers the question: given this background, what does it mean for something to be reliably re-identifiable by an external observer, as the thing that it is—it makes no metaphysical claims per se. See Ramstead, Sakthivadivel, & Friston (2022); Ramstead et al., (2023).
in line 173, it was unclear to me what "Here, G(π)" is.
It refers to the expected free energy of a policy. We have clarified in the text body.
Many thanks. We hope that these revisions are what you had in mind.
Reviewer 2 Report
Comments and Suggestions for Authors
The authors aim to scale Husserlian temporality, particularly protentions, to group intentionality using a category theory approach to the framework of active inference. At the outset, the authors neglect the first big “why.” Why is it interesting or important to model shared protentions in the first place? What will protentions, in particular, tell us? Husserl states that retention, primal impression, and protention form a unity, which is a form of passive synthesis, but here the authors mainly aim to understand protentions. The big “why” needs clarification. To make my review more concrete, I take the liberty to suggest the authors may want to understand how group temporality develops (and not emerges) despite individual temporalities.
To meet their aim, their first challenge is to establish a form of intersubjective temporality (see also Lanei M. Rodemeyer’s work) to shape the foundation for shared protentions. Much of the existing literature on particularly multiagent modeling, social cognition, and shared intentionality has been omitted (see for instance the work of; Siposova & Carpenter, 2019; Tollefsen, 2015; Vincini, 2023). Despite the rich literature, this challenge was only briefly touched upon in Section 2.2 and Section 4.0 (see below for concrete passages). Surely, the authors do not hold that shared protentions correspond exclusively to the predictions in the update function of the group model. This would correspond to saying that group dynamics only depend on the protentions we share, when in fact this is the problem of most group dynamics; the development depends on prediction we do not share due to our individual retentions and primal impressions. Perhaps the authors mean to argue that multiple agents can share the same trajectory (or have morphological similarities) in phase space, but this surely does not depend on protentions alone. I advise the authors to expand on this topic that is central to their aim. Particularly because it is necessary to know what they are modelling. This brings me to their second challenge.
The authors set out to model group dynamics (which, as a sidenote, is indeed possible with the existing work in hierarchical active inference) using category theory from where they intend to read off the protentions in the update function. First, analyzing the update function of a multiagent model would indeed allow a deeper understanding of how group temporality develops, however, the update function includes more than protentions. Features of the past (a priori likelihood, preferences, etc.) are included too. Second, if shared protentions can be read off the update function in a multiagent model, the authors should discuss possible complications in dealing with more than one timeline in the modelling as this suggests the existence of more than one protention at the time. If category theory allows dealing with time in a nontraditional way, they authors should clarify exactly how. Third, it is unclear why category theory is necessary when hierarchical active inference can model group dynamics using the original formalism.
Below you will find my review. As I am no expert in category theory, I have only made few comments. My concern is that the authors spend about half the paper introducing category theory and translating active inference to the new formalism instead of focusing on creating a solid ground in Husserlian temporality. Perhaps there are more than one paper here.

Author Response
The authors aim to scale Husserlian temporality, particularly protentions, to group intentionality using a category theory approach to the framework of active inference. At the outset, the authors neglect the first big “why.” Why is it interesting or important to model shared protentions in the first place? What will protentions, in particular, tell us? Husserl states that retention, primal impression, and protention form a unity, which is a form of passive synthesis, but here the authors mainly aim to understand protentions. The big “why” needs clarification. To make my review more concrete, I take the liberty to suggest the authors may want to understand how group temporality develops (and not emerges) despite individual temporalities.
We thank the reviewer for their very useful comments, which have greatly helped us to improve the paper.
Here we outline some broad, outline-level changes that we have made to the manuscript in response to the general comments made by the reviewer.
First, we now clarify the intended audience for this paper, which was lacking from the original manuscript, as the reviewer pointed out. We clarify that our intended audience is twofold. The primary intended audience is composed in part of phenomenologists who are interested in using contemporary mathematical approaches to generate formal models of the kinds of dynamic lived experiences that are captured by phenomenological descriptions. Our intended audience also comprises active inference modelers who have taken an interest in consciousness and phenomenological description.
We have added:
“This paper weaves together elements that might seem relevant to fairly disparate readerships: namely, Husserlian phenomenology, active inference modeling, and category theory. We clarify that our primary intended audience is threefold. The primary intended audience is composed in part of phenomenologists who are interested in using contemporary mathematical approaches to generate formal models of the kinds of dynamic lived experiences that are captured by phenomenological descriptions. This segment of our readership will likely intersect with proponents of the project to naturalize phenomenology (Ramstead, 2015; Petitot et al., 1999). Our target audience also comprises active inference modelers who have taken an interest in consciousness and phenomenological description.”
The reviewer’s point about the “why” question not being answered at the outset is a great one, which we have now set out to answer. We hope to have helped towards an answer to this by being clearer about our intended audience. We also clarify now why we appeal to Husserl specifically.
We have added:
“We appeal to Husserl’s phenomenological descriptions of inner time consciousness over those of other phenomenologists, e.g., de Beauvoir or Merleau-Ponty, for a number of key reasons. The first is practical: Husserl’s extensive body of work provides us with what are arguably the most comprehensive, rigorously conducted, and rich descriptions of first-person experience available in the phenomenological literature—which, in addition, are perhaps the most amenable to mathematical formalization, as Husserl himself attempted in a few key places. Although others in the phenomenological tradition, such as Heidegger, have proposed descriptions of time consciousness, we would argue that they are neither as descriptively rich as Husserl’s, nor as amenable to mathematization. Second, we chose Husserl’s phenomenological descriptions as a starting point because the recent project of computational phenomenology has already first been developed by formalizing Husserl’s descriptions of inner time consciousness using active inference modeling. So, we are choosing Husserl’s descriptions, in part, to build on and make the most of previous work. We should note that, following the tradition in naturalized phenomenology (e.g., Petitot et al., 1999), we propose to mainly use Husserl’s phenomenological descriptions to generate data to be explained using generative modeling—and bracket his antinaturalist philosophical commitments (see Ramstead, 2015, for a discussion).”
”
To meet their aim, their first challenge is to establish a form of intersubjective temporality (see also Lanei M. Rodemeyer’s work) to shape the foundation for shared protentions. Much of the existing literature on particularly multiagent modeling, social cognition, and shared intentionality has been omitted (see for instance the work of; Siposova & Carpenter, 2019; Tollefsen, 2015; Vincini, 2023). Despite the rich literature, this challenge was only briefly touched upon in Section 2.2 and Section 4.0 (see below for concrete passages).
This is a useful comment.
We have added a new subsection to address the existing literature on multiagent modeling, social cognition, and shared intentionality and relate it to our account.
We have added:
“There is a body of related work on shared intentionality, intersubjectivity, and joint/shared attention that is relevant in this context—see, e.g., Tomassello & Carpenter (2007), Tomassello (2014), Siposova and Carpenter (2019), and Vincini (2023). Joint attention is defined as the state in which two or more individuals focus on the same object or event. It involves mutual understanding and coordination of attention, underpinned by cognitive processes that enable individuals to recognize and follow each other's gaze or attentional focus, thus establishing a shared point of interest. Shared intentionality is the capacity to share psychological states, including intentions, beliefs, and goals, with others in order to coordinate.
These concepts have been developed in a Bayesian framework, which is quite germane to our own approach. Intersubjectivity here is seen as a consequence of agents conforming to the dependence structures harnessed in the same (or similar) generative models, and of the capacity of the agents to consider events and their sequentiality in a similar way, coordinating around them. For instance, Maisto, Donnarumma, and Pezzulo (2023) have argued that shared intentionality depends on agents having a shared representation of the joint goal and its context. This is a key aspect of intersubjectivity, as the agents maintain beliefs about a shared goal that they are trying to accomplish together. Agents continuously update their beliefs about the joint goal, which acts as a context based on observing each other's actions. This involves inferring the other's intentions and goals based on their movements. Through this kind of interactive inference, the agents align their representations and behaviors over time. This models the intersubjective alignment and "coupling" of representations that enables successful joint action in humans.
In a different study, Friedman, et. al (2021) investigated ant colony foraging behavior using an active inference model. The study models ant behavior in a T-maze paradigm, simulating how ants discover food sources and communicate these locations to the colony through pheromone trails. This behavior is formalized via active inference, emphasizing the ants' ability to make predictions and inferences about their environment based on sensory inputs and prior knowledge. This study also illustrates how shared representations and goals (e.g., locating food sources) facilitate coordinated actions within the ant colony. The ants' behavior, driven by hierarchical Bayesian inference, showcases a form of temporal integration where actions are predictive of the future, based on the anticipated positions of resources and the inferred intentions of other ants.
We draw partly from this work to justify our claim that the structure of temporality enables coordination, but we take a significant step beyond merely sharing similar predictive models. We suggest that agents do not just align their predictions of the future, but rather, also integrate the predictive models of others into their own. This integration reflects a deeper level of intersubjectivity, where the contents that are protended include anticipatory representations of other agents' future states of mind. Such integration is akin to a form of theory of mind, where understanding and predicting the mental states of others—including their intentions and future actions—are essential for complex social interactions. It also entails that the distinction between self and other gets blurrier as we move up a predictive hierarchy, explaining the co-constitutive nature of the self.
Surely, the authors do not hold that shared protentions correspond exclusively to the predictions in the update function of the group model. This would correspond to saying that group dynamics only depend on the protentions we share, when in fact this is the problem of most group dynamics; the development depends on prediction we do not share due to our individual retentions and primal impressions. Perhaps the authors mean to argue that multiple agents can share the same trajectory (or have morphological similarities) in phase space, but this surely does not depend on protentions alone. I advise the authors to expand on this topic that is central to their aim. Particularly because it is necessary to know what they are modelling. This brings me to their second challenge.
We agree with the reviewer here—and realize that the manner that we chose to express ourselves could have been much clearer, and truer to Husserl’s.
We have added a section on the sedimentation of contents and on shared protentional/retentional contents:
“We now make a terminological clarification. This paper concerns what we will call, in a somewhat idiosyncratic fashion, “shared protentions.” It should be noted that for Husserl, in some sense, the structure of time consciousness, as an intentional relation between primary content, retention, and protentions, are shared by all conscious subjects: the structure of inner time consciousness is invariant and identically the same for all conscious subjects. That is, according to Husserl, inner time consciousness must conform to this structure, by virtue of being the experience of the type: inner time consciousness.
“The co-constitution of the lived world (Lebenswelt) has been discussed extensively by Husserl, and studied under the rubric of intersubjectivity by phenomenologists of various sorts. Husserl distinguishes between the empirical and transcendental self or ego. For Husserl, the self generically functions as one of two “poles” of conscious experience (the “act-pole” and the “object-pole”). The transcendental self is, on the one hand, a kind of abstract structure that any conscious experience contains intrinsically. The self is also a thing that appears to this transcendental self: this is the empirical or personal self, which also has a temporal structure, and accumulates experiences, forming habits. This empirical self is not just a static center of acts but evolves over time, integrating past acts into a cohesive identity. In his later writings, in particular in the fifth Cartesian Meditation (Husserl, 2013), The Crisis of European Sciences and Transcendental Philosophy (Husserl, 1936), and the extensive notes On Intersubjectivity, Husserl extendshis analysis of the self, considered through the lens of intersubjectivity.
On Husserl’s account, the empirical self is co-constituted with others, where past experiences become integrated into the self. Since these experiences almost always involve other selves, their perspectives are thus integrated to the self. The self is thus a collective achievement, that emerges from a community of selves (Heinamaa, 2021). The self realizes its constitutive role only within a network of intersubjective relations to other selves, implying an intrinsic connection between individual consciousnesses. The self is co-constructed through this intersubjective framework, where each self intentionally carries within itself the presence of other selves, thereby forming a deep, communicative relationship that establishes the full sense of the world. This communal interaction and the acknowledgment of each other's subjective experiences contribute to the continuous generation and reformation of the self, emphasizing a dynamic, interrelated construction of identity and understanding (Heinamaa, 2021). Our individual temporal experiences, the immediate experience of the now, retention, and protention, are not closed off to ourselves. They are open and connected to the temporal experiences of others. This openness allows for a shared temporal framework or "world-time" that encompasses not only our own temporal flow but also the temporal experiences of others, making our individual sense of time inherently intersubjective (Rodemeyer, 2006).
In the flow of time consciousness, the contents of experience become “formatted” by retention and protention, as pure forms of time consciousness, as they well up. It is these sedimented contents to which we refer—as shorthand—as “shared retentions” and “shared protentions”; in the sense that the content is shared between conscious subjects.
Previous work on modeling Husserlian time consciousness with active inference has associated these contents that are retained and protended by the pure protentional and retentional structure of inner time consciousness to the implicit knowledge that is encoded in the parameters of a generative model, which captures the formal structure of inference. This is to be contrasted with the kind of online, contentful inference that corresponds to posterior state estimation in real time—i.e., with explicit predictions about the future. Thus, we should note that protentions are not equivalent to explicit predictions (neither in Husserl’s sense of formal structure, nor in our sense of retained or protended contents). The point we are making is that shared intentionality depends on shared sedimented content; that is, in addition to having in common the generic form of inner time consciousness, shared protentional content is necessary for shared intentionality.”
The authors set out to model group dynamics (which, as a sidenote, is indeed possible with the existing work in hierarchical active inference) using category theory from where they intend to read off the protentions in the update function.
This is an interesting point, but we disagree. What is not possible at present in the classic active inference framework is to model how each conscious subject specifically experiences their lifeworld—that is, how things ‘look from their perspective.’ Perhaps more importantly, the classic framework formally concentrates on the interaction of a single agent with its environment. Our category theoretic generalization allows us to confront both of these deficits, describing first agents that believe themselves to be in a social context and then formalizing how their compatible beliefs yield shared protentions. Doing this mathematically rigorously is not possible in the classic framework.
First, analyzing the update function of a multiagent model would indeed allow a deeper understanding of how group temporality develops, however, the update function includes more than protentions. Features of the past (a priori likelihood, preferences, etc.) are included too.
Agreed. We now point this out, see additions outlined just above.
Second, if shared protentions can be read off the update function in a multiagent model, the authors should discuss possible complications in dealing with more than one timeline in the modelling as this suggests the existence of more than one protention at the time. If category theory allows dealing with time in a nontraditional way, they authors should clarify exactly how. Third, it is unclear why category theory is necessary when hierarchical active inference can model group dynamics using the original formalism.
It is unclear to us why multi-timeline modeling would be useful.
In category theory, a category is just a collection of objects and their relations (i.e., maps between objects), along with composition operations that are governed by associativity and identity rules. Category theory provides us with a way of expressing generic mathematical structure, in a way that enables us to identify common structures in formalisms. In this technical sense, category theory is absolutely general and applies to every area of mathematical inquiry, revealing the underlying structures of the formalisms being studied. The reason we mention this is that, generally speaking, if agents can be described as having or conforming to any mathematical structure at all, then that structure can be studied through the lens of compositionality using category theory.
Category theory is useful here because it provides us with precisely the kinds of tools required to formalize the notion of the world that we perceive, or lifeworld. This kind of perception surely must have a mathematical structure.
Below you will find my review. As I am no expert in category theory, I have only made few comments. My concern is that the authors spend about half the paper introducing category theory and translating active inference to the new formalism instead of focusing on creating a solid ground in Husserlian temporality. Perhaps there are more than one paper here.
Major issues.
- “Shared protentions”. Throughout the paper, the authors loosely use concepts as “shared goals” and “group intentionality” as synonymous with “shared protentions.” Traditionally, the three temporal intentionalities (retention, primal impression, and protention) are for apprehension, i.e., passive synthesis of intentionality, and thus not operational in the domain of goals and practicalities.
Great point. See response above.
The temporal structure allows for the experience of contents, but does not determine them. Protention is a fundamental aspect of the intentional structure of consciousness, relevant to goal-seeking behavior, but not a goal in itself. We have protentions with or without goals, alone or with others.
We agree with all this. Again, please see responses above.
Even though protentions possess a quality of "striving" just like goal-seeking behavior, they maintain an openness to a horizon of possibilities, which is more of an exploratory nature (Hua XI, 73). And keep in mind, Husserl emphasized that the striving nature of protention is a form of passive directedness, i.e., "passive intentionality" (Hua XI, 76). This does not support the idea of relating protentions directly to goal-directed behavior. My hunch is that protentions have been understood as predictions in Bayesian terms. Do the authors believe that “protentions” mean exactly the same as “predictions” in Bayesian inference? To be sure, I have highlighted the problematic places.
We should have said that shared protentions are a necessary condition for the emergence of shared intentionality; see text added above.
o P. 4, line 140-142: “A shared protention might refer to isomorphic protentions, implicitly shared among agents that are not in direct interaction or communication.” Perhaps the choice of word is unintentional (pun unintended), but isomorphism refers to structural relations. Protentions are part of a larger (inseparable) structure of temporality (see Gallagher & Zahavi, 2012), but the structure remains the same. The structural relationship between retention, primal impression, and protention is invariant. The content and pace of the flow may differ, but all protentions (in healthy bodies) already share the same position in a grander temporal structure.
This is a very astute point. As indicated above, we have rewritten sections of the manuscript to reflect the difference between the invariant structure of time consciousness (which is de facto shared by all conscious subjects) and what we were calling, idiosyncratically, shared protections—which may be better understood as shared protended contents. Again, please see added text above.
o P. 4, line 143-145: “Another sense of shared protention might be a belief that can be ascribed to an ensemble per se, for example, the protentions of a social group might be understood as arising from interactions between individual agents of a common cultural background, each with their own, distinct, individual-level protentions.” The ensemble version of shared protentions is vague and effectively only reveals how they emerge—not what they are. Please unpack what shared protentions are when they depend on social interaction and how is this different from the above.
See additions above.
o To resolve this, I believe the authors need to unfold the definition of protention that speaks to their non-traditional view. The questions are (i) to what extent is a protention a subjective construct, and (ii) to what extent should we distinguish between intentionality as consciousness and directedness. Protentions are traditionally the latter. Would the authors say that two geographically distant persons share protentions in reaching out for a cookie at the same time? How exactly do these protentions interact (since interaction is a prerequisite for sharing)?
- The value of using category theory is unclear as the conclusions could have been reached using hierarchical active inference. This would be clear if the approach was evidenced to be computationally less expensive, but this was not the case. Instead, the focus has been just as much on converting features of the active inference into category theory as the phenomenological mapping and analysis. This obscures the aim of the paper.
We agree that our use of category theory is poorly motivated in the submitted version of the paper. As discussed above, we use category theory for two main reasons: firstly, because it allows us to be precise about the essential structure of active inference systems, and therefore to generalize them to new contexts (such as the present one); and secondly, because it is inherently relational and therefore uniquely well allows us to articulate what it means to experience a world from a given perspective, and to share that world with peers.
o Additionally, the authors end on a note regarding topos as a shared universe. Though I am not an expert on category theory, however, it seems to me that the authors have demonstrated that multiple objects/agents (which are eventually sets) can have overlapping world models and thereby share the same topos category. I may be wrong, but does this not mean that the multiple agents share common properties or relationships within the topos? If this is the case, then stating that the two agents share a universe is a stretch, when really, they share properties and relationships, i.e. they are under the same category. According to my reading, as non-expert in category theory, I cannot determine whether the results have been clearly presented.
A topos is simply a “structured space”; inherently, the internal world of an agent is such a space, and therefore may be modeled as a topos. More importantly, topos theory provides tools that allow us to measure when two toposes may be “glued together” (assuming they somehow overlap). Our proposal is simply to use these tools to analyze the world as understood to be shared by a group of agents, and thus to come to a mathematically precise understanding of their social phenomenology.
Minor issues.
- P. 1, line 27, & p. 5, line 189. To my knowledge, Yoshimi’s book on Husserlian phenomenology as a unifying interpretation is not between phenomenology and active inference, but of a dynamical systems approach without any Bayesian approximation methodology. Active inference is not exclusive in using previous information/experience to underwrite expectations for a better understanding of the world. This is also observed in various neural networks, multivariate autoregressors and, broadly speaking, any form of regression modeling. Please, correct the reference.
Corrected.
- P. 5, line 173: Please correct the discrete version; G(pi), to the continuous version; G(u).
This is incorrect: u denotes actions, not policies.
- P. 5, line 193-210: This whole paragraph is precisely what Bogotá & Djebbara (2023) set out to analyze which would only back up your claim.
We now cite Bogotá & Djebbara (2023)
- P. 8, line 287-288: Something is off with this sentence. Please revise.
Corrected.
References
Bogotá, J. D., & Djebbara, Z. (2023). Time-consciousness in computational phenomenology: a temporal analysis of active inference. Neuroscience of Consciousness, 9(1), 1–12.
Gallagher, S., & Zahavi, Dan. (2012). The phenomenological mind (2nd ed.). Routledge.
Hua XI: Husserl, E. (1966). Analyse des Zeitbewusstseins, in Vorlesungen zur Phänomenologie des inneren Zeitbewusstseins, Den Haag, Nijhoff, pp. 19-72.
Siposova, B., & Carpenter, M. (2019). A new look at joint attention and common knowledge. Cognition, 189, 260–274.
Tollefsen, D. P. (2015). Groups as agents. John Wiley & Sons.
Vincini, S. (2023). Taking the mystery away from shared intentionality: The straightforward view and its empirical implications. Frontiers in Psychology, 14, 1068404.
Many thanks. We hope that these revisions are what you had in mind.
Reviewer 3 Report
Comments and Suggestions for Authors
To get the only concrete thing I have to say out of the way I noticed that in table 1 we have the variable "c", and I did not see where "c" was defined. Also in the fifth cell of Table 1 there is a missing "t" . It says P(O_{+1}... when it should be P(O{t+1}... And lines 215 and 216 are missing some words when they say:
... the data infers that are inferred from the sensory input ...
Now, to my opinion on the text.
Since this issue of the journal is a festschrift for Karl Friston, and Karl Friston is an author on the article it seems hard to say much beyond accept, but I will offer several comments that the authors together can address or ignore as they prefer.
I don't know who is the target audience for this. The three sections of the article are fairly independent. None of them depend on the other to persuade a reader that their take on the problem is the correct one. And given the complexity of each domain it is unlikely that anyone unfamiliar with, say active inference, will gain any new insights from struggling with Husserl and his neologisms. The same worry applies to the category theory section. Even basic category theory is a field saturated with nomenclature unfamiliar to the non-specialist. The category theory offered here is particularly cutting edge and is unlikely to be familiar to even most practicing category theorists. If the authors had a particular reader in mind maybe it would become clearer how to rework the treatment to make sure all sections are accessible to this modern Descartes.
It also isn't clear if there is any necessity to linking these topics here. Some of this limited development is almost certainly due to what I suspect was a space constraint on the submissions, but if that is the case the article's clarity and reach would be improved by considerably scaling back the treatment. I would advise just giving us the thumbnail sketch and using the references to guide those who are interested to where they can pursue the topics in more depth. Questions I had when reading the article were some of the following (offered in case they help the authors to decide whether or how to re-work the manuscript):
1. Why Husserl's treatment of time? The need to have retrospection and prospection seems pretty obvious. The paper doesn't make clear what it is specifically about Husserl's approach that makes it better than any other for the modern reader. Is Husserl being used substantively here or just as a convenient way to introduce the need to blend past, present and future into a sliding window of awareness?
2. Maybe the direction of influence is supposed to be in the other direction? We are imagining most readers to be phenomenologists or at least philosophers and we want to use the mapping between active inference and the different past, present, and future features of time consciousness to lend support for Husserl's take? If so, then a more discriminating argument should be made. Why is the fit to Husserl better than any other take on time consciousness?
3. What is category theory buying us here? In general, category theory can hint at similar structures and roles in different domains, some applied and some pure. But here we only get a glimpse of the building of the edifice without the sharing of any new insights. What do we gain from viewing protensions and free energy ideas in this framework that were not already apparent from the metaphorical treatment that preceded the categorical framing? What new directions could we take once we have seen the categorical framing that might not have occurred to an active inference expert before?
4. Why laden active inference, which is already seen as an impenetrable topic by many, with two new domains of deeply difficult and jargon laden areas? There must be a commensurate gain in insight to warrant the reader slogging through all the verbiage. And why do we have to use the verbiage? What does the use of Husserl's nomenclature buy us? Why protension for instance? I am easily persuaded that if the majority of readers are to be phenomenologists then the standard vocabulary of the field makes sense, but, again, do we expect them to be wading through the next two sections, and if so, give them a reason why?
5. Only say what you have the space to say clearly. This is a short paper. A more modest treatment and a more direct vocabulary would help immensely. Here is what I take away (of course it may be wrong, but maybe that will help the authors see where they need more clarity) mixed with my suggestions for scaling back:
1. People often act in concert thus communicating some common expectations about the future.
2. A vocabulary for how it is that we can in the present moment also have a linkage to the past and a prediction for the future is helpful, and we may stumble onto new perspectives if we can take this vocabulary from others who have thought deeply about this issue before. Husserl is one such thinker, but his philosophy is highly complex and debated so we will only use a minimal set of terms while pointing the reader to other literature for deeper treatments.
3. When people act it means they observe, think and decide. We would like to have a unified account of that and active inference is offered. Since we are interested in the social commonalities, maybe even coordination of different minds, we can try to use our insights from number two and our goal of describing number 1 as we seek to emphasize the parts of active inference that are the best match.
4. And we might wonder if all the above, to the extent it holds together, is at all natural or is it ad-hoc and idiosyncratic? Well, this is the sort of thing math is good at. Helping us be concrete to clearly see the consequences of our particular frameworks. Category theory is a good player here, and while we cannot develop our ideas fully (that is one cannot compress a 300 page archive paper down to five pages), we can illustrate the direction of such work by, for example, emphasizing the notion of category and functors to hint at the way that well developed, separate, areas can be seen to exhibit the same structure, and how the discovery or description of the functor makes that shared structure manifest.
Maybe, and it is a soft maybe, a simplified treatment of the above might be read.
Author Response
To get the only concrete thing I have to say out of the way I noticed that in table 1 we have the variable "c", and I did not see where "c" was defined. Also in the fifth cell of Table 1 there is a missing "t" . It says P(O_{+1}... when it should be P(O{t+1}... And lines 215 and 216 are missing some words when they say:
... the data infers that are inferred from the sensory input …
Thanks for catching this.
We notice that the variable μ is similarly undefined.
We have added explanations for c and μ
We have changed P(O_{+1} to P(O_{t+1}
We have amended the sentence to add the missing words
Now, to my opinion on the text.
Since this issue of the journal is a festschrift for Karl Friston, and Karl Friston is an author on the article it seems hard to say much beyond accept, but I will offer several comments that the authors together can address or ignore as they prefer.
I don't know who is the target audience for this. The three sections of the article are fairly independent. None of them depend on the other to persuade a reader that their take on the problem is the correct one. And given the complexity of each domain it is unlikely that anyone unfamiliar with, say active inference, will gain any new insights from struggling with Husserl and his neologisms. The same worry applies to the category theory section. Even basic category theory is a field saturated with nomenclature unfamiliar to the non-specialist. The category theory offered here is particularly cutting edge and is unlikely to be familiar to even most practicing category theorists. If the authors had a particular reader in mind maybe it would become clearer how to rework the treatment to make sure all sections are accessible to this modern Descartes.
We are grateful for this remark. We now clarify that the intended audience for this paper is phenomenological thinkers interested in using formal approaches to help them model their descriptions of phenomenological experience.
We have added:
“This paper weaves together elements that might seem relevant to fairly disparate readerships: namely, Husserlian phenomenology, active inference modeling, and category theory. We clarify that our primary intended audience is threefold. The primary intended audience is composed in part of phenomenologists who are interested in using contemporary mathematical approaches to generate formal models of the kinds of dynamic lived experiences that are captured by phenomenological descriptions. This segment of our readership will likely intersect with proponents of the project to naturalize phenomenology (Ramstead, 2015; Petitot et al., 1999). Our target audience also comprises active inference modelers who have taken an interest in consciousness and phenomenological description.”
It also isn't clear if there is any necessity to linking these topics here. Some of this limited development is almost certainly due to what I suspect was a space constraint on the submissions, but if that is the case the article's clarity and reach would be improved by considerably scaling back the treatment. I would advise just giving us the thumbnail sketch and using the references to guide those who are interested to where they can pursue the topics in more depth.
This is a good suggestion. We have decided to expand the paper a bit to do justice to the complexity of the arguments. We have added some sign posts throughout to indicate opportunities for further reading to the reader.
Questions I had when reading the article were some of the following (offered in case they help the authors to decide whether or how to re-work the manuscript):
- Why Husserl's treatment of time? The need to have retrospection and prospection seems pretty obvious. The paper doesn't make clear what it is specifically about Husserl's approach that makes it better than any other for the modern reader. Is Husserl being used substantively here or just as a convenient way to introduce the need to blend past, present and future into a sliding window of awareness?
Husserl’s descriptions are sufficiently precise that they lend themselves well to mathematical formalization. Husserl himself tried to do this.
- Maybe the direction of influence is supposed to be in the other direction? We are imagining most readers to be phenomenologists or at least philosophers and we want to use the mapping between active inference and the different past, present, and future features of time consciousness to lend support for Husserl's take? If so, then a more discriminating argument should be made. Why is the fit to Husserl better than any other take on time consciousness?
Correct, we have clarified this in a section discussing our intended audience.
We have added:
“This paper weaves together elements that might seem relevant to fairly disparate readerships: namely, Husserlian phenomenology, active inference modeling, and category theory. We clarify that our primary intended audience is threefold. The primary intended audience is composed in part of phenomenologists who are interested in using contemporary mathematical approaches to generate formal models of the kinds of dynamic lived experiences that are captured by phenomenological descriptions. This segment of our readership will likely intersect with proponents of the project to naturalize phenomenology (Ramstead, 2015; Petitot et al., 1999). Our target audience also comprises active inference modelers who have taken an interest in consciousness and phenomenological description.”
- What is category theory buying us here? In general, category theory can hint at similar structures and roles in different domains, some applied and some pure. But here we only get a glimpse of the building of the edifice without the sharing of any new insights. What do we gain from viewing protensions and free energy ideas in this framework that were not already apparent from the metaphorical treatment that preceded the categorical framing? What new directions could we take once we have seen the categorical framing that might not have occurred to an active inference expert before?
We agree that our use of category theory is poorly motivated in the submitted version of the paper. As discussed above, we use category theory for two main reasons: firstly, because it allows us to be precise about the essential structure of active inference systems, and therefore to generalize them to new contexts (such as the present one); and secondly, because it is inherently relational and therefore uniquely well allows us to articulate what it means to experience a world from a given perspective, and to share that world with peers.
We have added two subsections to §5 to make clearer the connection between the classical active inference framework and the category-theoretic presentation.
- Why laden active inference, which is already seen as an impenetrable topic by many, with two new domains of deeply difficult and jargon laden areas? There must be a commensurate gain in insight to warrant the reader slogging through all the verbiage. And why do we have to use the verbiage? What does the use of Husserl's nomenclature buy us? Why protension for instance? I am easily persuaded that if the majority of readers are to be phenomenologists then the standard vocabulary of the field makes sense, but, again, do we expect them to be wading through the next two sections, and if so, give them a reason why?
We clarify now why we appeal to Husserl specifically.
We have added:
“We appeal to Husserl’s phenomenological descriptions of inner time consciousness over those of other phenomenologists, e.g., de Beauvoir or Merleau-Ponty, for a number of key reasons. The first is practical: Husserl’s extensive body of work provides us with what are arguably the most comprehensive, rigorously conducted, and rich descriptions of first-person experience available in the phenomenological literature—which, in addition, are perhaps the most amenable to mathematical formalization, as Husserl himself attempted in a few key places. Although others in the phenomenological tradition, such as Heidegger, have proposed descriptions of time consciousness, we would argue that they are neither as descriptively rich as Husserl’s, nor as amenable to mathematization. Second, we chose Husserl’s phenomenological descriptions as a starting point because the recent project of computational phenomenology has already first been developed by formalizing Husserl’s descriptions of inner time consciousness using active inference modeling. So, we are choosing Husserl’s descriptions, in part, to build on and make the most of previous work. We should note that, following the tradition in naturalized phenomenology (e.g., Petitot et al., 1999), we propose to mainly use Husserl’s phenomenological descriptions to generate data to be explained using generative modeling—and bracket his antinaturalist philosophical commitments (see Ramstead, 2015, for a discussion).”
- Only say what you have the space to say clearly. This is a short paper. A more modest treatment and a more direct vocabulary would help immensely. Here is what I take away (of course it may be wrong, but maybe that will help the authors see where they need more clarity) mixed with my suggestions for scaling back:
- People often act in concert thus communicating some common expectations about the future.
- A vocabulary for how it is that we can in the present moment also have a linkage to the past and a prediction for the future is helpful, and we may stumble onto new perspectives if we can take this vocabulary from others who have thought deeply about this issue before. Husserl is one such thinker, but his philosophy is highly complex and debated so we will only use a minimal set of terms while pointing the reader to other literature for deeper treatments.
Following Petitot et al (1999) we propose to mainly use Husserl’s phenomenological descriptions and bracket his philosophy. See remarks just above.
- When people act it means they observe, think and decide. We would like to have a unified account of that and active inference is offered. Since we are interested in the social commonalities, maybe even coordination of different minds, we can try to use our insights from number two and our goal of describing number 1 as we seek to emphasize the parts of active inference that are the best match.
- And we might wonder if all the above, to the extent it holds together, is at all natural or is it ad-hoc and idiosyncratic? Well, this is the sort of thing math is good at. Helping us be concrete to clearly see the consequences of our particular frameworks. Category theory is a good player here, and while we cannot develop our ideas fully (that is one cannot compress a 300 page archive paper down to five pages), we can illustrate the direction of such work by, for example, emphasizing the notion of category and functors to hint at the way that well developed, separate, areas can be seen to exhibit the same structure, and how the discovery or description of the functor makes that shared structure manifest.
Maybe, and it is a soft maybe, a simplified treatment of the above might be read.
Many thanks. We hope that these revisions are what you had in mind.
Round 2
Reviewer 2 Report
Comments and Suggestions for Authors
All my points have successfully been addressed by the authors.